# Knockout of the orphan membrane transporter *Slc22a23* leads to a lean and hyperactive phenotype with a small hippocampal volume

**Yasuhiro Uchimura**[1]*, **Kodai Hino**[1†], **Kosuke Hattori**[2], **Yoshinori Kubo**[1], **Airi Owada**[1], **Tomoko Kimura**[1], **Lucia Sugawara**[3], **Shinji Kume**[3], **Jean-Pierre Bellier**[4¤], **Daijiro Yanagisawa**[4], **Akihiko Shiino**[4], **Takahisa Nakayama**[5], **Yataro Daigo**[6], **Tomoji Mashimo**[2], **Jun Udagawa**[1]*

1 Division of Anatomy and Cell Biology, Department of Anatomy, Shiga University of Medical Science, Otsu, Shiga, Japan, 2 Division of Animal Genetics, Laboratory Animal Research Center, Institute of Medical Science, The University of Tokyo, Tokyo, Japan, 3 Department of Medicine, Shiga University of Medical Science, Otsu, Shiga, Japan, 4 Molecular Neuroscience Research Center, Shiga University of Medical Science, Otsu, Shiga, Japan, 5 Division of Human Pathology, Shiga University of Medical Science, Otsu, Shiga, Japan, 6 Department of Medical Oncology, Cancer Center and Center for Advanced Medicine against Cancer, Shiga University of Medical Science, Otsu, Shiga, Japan

† Deceased.
¤ Current address: Department of Neurology, Brigham and Women's Hospital/Harvard Medical School, Cambridge, MA, United States of America
* uchimura@belle.shiga-med.ac.jp (YU); udagawa@belle.shiga-med.ac.jp (JU)

**Data Availability Statement:** All relevant data are within the manuscript and its Supporting Information files.

## Abstract

Epidemiological studies suggest that poor nutrition during pregnancy predisposes offspring to the development of lifestyle-related noncommunicable diseases and psychiatric disorders later in life. However, the molecular mechanisms underlying this predisposition are not well understood. In our previous study, using rats as model animals, we showed that behavioral impairments are induced by prenatal undernutrition. In this study, we identified solute carrier 22 family member 23 (*Slc22a23*) as a gene that is irreversibly upregulated in the rat brain by undernutrition during fetal development. Because the substrate of the SLC22A23 transporter has not yet been identified and the biological role of the *Slc22a23* gene *in vivo* is not fully understood, we generated pan-*Slc22a23* knockout rats and examined their phenotype in detail. The *Slc22a23* knockout rats showed a lean phenotype, an increase in spontaneous locomotion, and improved endurance, indicating that they are not overweight and are even healthier in an *ad libitum* feeding environment. However, the knockout rats had reduced hippocampal volume, and the behavioral analysis suggested that they may have impaired cognitive function regarding novel objects.

## Introduction

The developmental origins of health and disease (DOHaD) hypothesis has been proposed [1–3] because epidemiological studies have shown that nutritional stress during fetal development

**Funding:** This work was supported by Japan Society for the Promotion of Science (JSPS; https://www.jsps.go.jp/) - JP 19K08274 [YU], JP 16H06277 / JP 22H04923 [YD], and JP 16H06276 [TM]. The funders had no role in the study design, data collection and analysis, decision to publish, or manuscript preparation.

**Competing interests:** The authors have declared that no competing interests exist.

is significantly correlated with the onset of lifestyle-related non-communicable diseases (NCDs) such as obesity and type 2 diabetes and psychiatric disorders such as schizophrenia in adulthood [4, 5]. Although several studies have been reported [6, 7], the molecular mechanisms underlying the phenotypic changes associated with the DOHaD are still largely unknown. In this study, we found that the transcript of the solute carrier 22 family member 23 (*Slc22a23*) gene is irreversibly upregulated in the rat brain by undernutritional stress during fetal development.

The *Slc22a23* gene has previously been shown to have a high mRNA level in neurons in the brain and to exhibit homology to membrane cation transporters [8, 9]. In contrast, the gene with the highest homology to *Slc22a23* is *Slc22a17*, which has been reported as lipocalin 24p2 (lipocalin 2), a transporter of iron ions into cells [10], suggesting that SLC22A23 may function as a membrane ion transporter. However, based on sequence homology, SLC22A23 was classified into a subgroup of organic anion transporters in a recent review [11]. Thus, the identification of SLC transporter substrates remains challenging, and the biological role of the *Slc22a23* gene *in vivo* is not yet fully understood. In this study, we generated *Slc22a23* knockout rats and analyzed their phenotype in detail to elucidate the biological role of *Slc22a23 in vivo*.

## Materials and methods

### RNA array

Gestational day (GD) 10.5 embryos (n = 3 in each group) were embedded in an optimal cutting temperature compound (Leica Microsystems, Wetzlar, Germany) and were snap-frozen in liquid nitrogen as previously described [12]. Sections of 10 μm thickness were cut and fixed on adhesive membrane-coated microscope slides (Molecular Machines and Industries GmbH, Eching, Germany). Embryonic forebrain microdissections were isolated from the 10-μm thick sections using a laser microdissection system (Molecular Machines and Industries GmbH). mRNA samples were extracted from the microdissections and converted to cDNA. After Cy3-labeling, the cDNA samples were hybridized to the SurePrint G3 Rat GE3 8x60K microarray (Agilent Technologies, Santa Clara, CA, US). The sumz function in the metap package [13] in R (version 4.3.2) was used for meta-analysis of the p-value provided by the microarray analysis.

### Quantitative PCR

RNA was extracted from the frontal lobes of 9-week-old male rats and converted into cDNA. Real-time PCR analysis was performed using SYBR Premix Ex TaqII polymerase (Takara Bio, Shiga, Japan) on a LightCycler 480 system (Roche, Basel, Switzerland). The following primers were used for real-time PCR analysis: 5'-CGAGACCGATGTATATGCTTGC and 5'-GTCCAGATGATTCAGAGCTCCA for beta-2 microglobulin, 5'-TTCTGGACCAGCCCAACTTC and 5'-CAGGAGGCAGTGGAGAAAGG for *Slc22a23*, and 5'-GGTATCCCTTTGGCTTTCACA and 5'-GCCCAGCTTCCTCTTCAATCA for *Gnrh1*.

### Generation of *Slc22a23* knockout rats

First, we generated W-floxed-*Slc22a23* rats, a transgenic rat strain in which two loxP sequences were inserted into the introns flanking exon 4 of the *Slc22a23* gene. The W-floxed-*Slc22a23* strain was generated using the CRISPR-Cas system [14]. Then, to obtain *Slc22a23* knockout rats, the W-floxed-*Slc22a23* strain was crossed with the W-Tg(CAG-cre)81Jmsk strain, which expresses Cre recombinase under the CAG promoter [15]. The W-Tg(CAG-cre)81Jmsk strain was obtained from the National BioResource Project for the Rat in Japan (NBRP).

## Genotyping

After ear-punching for individual identification, the tail tips of rats were isolated and digested in a digestion buffer (50 mM Tris pH8, 100 mM EDTA, 100 mM NaCl, 1% SDS) containing 0.2 mg/ml Proteinase K (Wako, Osaka, Japan) at 55˚C until complete digestion. After separation using phenol:chloroform:isoamyl alcohol (25:24:1, Nacalai tesque, Kyoto, Japan), genomic DNA was precipitated from the supernatants with isopropanol. Then, each pellet was resuspended in TE buffer (10 mM Tris pH 8, 1 mM EDTA). The genomic DNA solutions were incubated at 95˚C for 10 minutes to completely inactivate any remaining Proteinase K activity. KOD One PCR Master Mix blue (TOYOBO, Osaka, Japan) was used for PCR genotyping. A primer set (F1: 5'-AACAGGTTCTTGTCAACAGG and R1: 5'-GAGGGCCTCATTTCACATTA) was used to amplify the floxed site of the *Slc22a23* gene, producing 1045 bp (wild-type), 1113 bp (floxed), and 622 bp (knockout) fragments. A primer set (5'-GCTGGTTGTTGTGCTGTCT CATC and 5'-ACCATTGCCCCTGTTTCACTATC) was used to genotype the Cre recombinase gene, and the amplified fragment was approximately 1.2 kb.

## Generation of a C2C12 cell line stably expressing SLC22A23

A hemagglutinin (HA) tag was fused to the C-terminus of the *Slc22a23* gene by PCR amplification using a forward primer 5'-AAAC AGATCT GGATCC gcc acc ATG GCG ATA GAC CGG CGG CGC GAG (BglII-BamHI-Kozak-*Slc22a23*) and a reverse primer 5'-AAAC GTCGAC CTA ggc ata gtc agg cac gtc ata agg ata CTCGAG GAATTC CAT GGT CTT CAT GCC ATT GGC AGT G (SalI-stop-HA-XhoI-EcoRI-*Slc22a23*). Then, the fragment was subcloned into a retroviral vector plasmid, pMXs-IRES-Puro [16], using BamHI/BglII ligation and SalI/XhoI ligation. To obtain murine ecotropic non-self-replicable retroviral particles, the pMXs-*Slc22a23*-HA-IRES-Puro plasmid was transfected into PlatE cells [17] using FuGENE HD transfection reagent (Promega, Madison, WI, US). The supernatant of the PlatE cell culture was mixed with polybrene (final 4 μg/ml) (Nacalai tesque) and added to the murine myoblast cell line C2C12. One day after retroviral transfection, C2C12 cells were selected using 2 μg/ml puromycin (InvivoGen, San Diego, CA, US). After 10 days in culture, puromycin-resistant clones were isolated from 96-well plates. Expression of HA-fused SLC22A23 protein was confirmed by western blotting using an anti-HA antibody (1:1000) (clone 3F10; Roche Cat# 11867423001, RRID:AB_390918) and HRP-conjugated anti-rat IgG (1:15000) (Cytiva Cat# NA935, RRID:AB_772207; Marlborough, MA, US).

## Production of rabbit antiserum against SLC22A23

For antigen preparation, a region of *Slc22a23* (amino acids 622–689) was amplified by PCR using the forward primer 5'-AAAC AGATCT GAA AAC CTG TAT TTC CAG TGC GGATCC CCT GAG AGC AGG GAC CAG (BglII-TEV cleavage site (ENLYFQ ▼C)-BamHI-*Slc22a23* forward) and the reverse primer 5'-AAAC GAATTC CTA CAT GGT CTT CAT GCC (EcoRI-stop-*Slc22a23* reverse). The cDNA was subcloned into a pGEX 4T-1 plasmid (Cytiva) using BamHI/BglII ligation and an EcoRI site. Rosetta-gami^TM 2(DE3) (Novagen cat# 71351; Darmstadt, Germany) was transformed with the pGEX-4T-1 plasmid. The transformed bacteria were cultured in LB medium with ampicillin and chloramphenicol selection. After centrifugation, the bacteria were disrupted by sonication in a lysis buffer (phosphate-buffered saline, 10 mM EDTA, 0.1% TritonX-100, 1 mM DTT, and protease inhibitors (0.5 mM ABSF, 0.15 mM aprotinin, 0.5 mM E-64, and 1 mM leupeptin (Nacalai tesque))). After centrifugation, glutathione *S*-transferase (GST) fusion proteins were purified from the bacterial supernatant using Glutathione Sepharose^TM 4B (Cytiva). SLC22A23 (amino acids 622–689) was cleaved

from GST using TEV protease. The resulting peptide was N-CGSPESRDQNLPENIANGEHYT RQPLLSHKKGEQPLLLTNAELKDYSGLHDVAAVGDGLSEGATANGMKTM. The cysteine residue at the N-terminus of the peptide was used for carrier protein conjugation. Three types of carrier proteins were used in this study: maleimide-activated keyhole limpet hemocyanin (KLH) (PEPTIDE INSTITUTE, Osaka, Japan), maleimide-activated ovalbumin (made in-house), and maleimide-activated blue carrier protein (Thermo Fisher Scientific, Waltham, MA, US). The KLH-conjugated peptide in complete Freund's adjuvant (BD, Franklin Lakes, NJ, US) was injected subcutaneously into a 9-week-old female rabbit (Slc:NZW; Japan SLC, Shizuoka, Japan). Six further rounds of immunization were performed every 2 weeks. KLH-conjugated antigen in TiterMax Gold (TiterMax USA, Norcross, GA, US) was injected at the second and third immunizations. Ova-conjugated antigen in incomplete Freund's adjuvant (BD) was injected at the fourth and fifth immunizations. Blue carrier protein-conjugated antigen in incomplete Freund's adjuvant was injected at the sixth and seventh immunizations. One week after the seventh immunization, the rabbit was anesthetized by 4% isoflurane inhalation (VIATRIS, Tokyo, Japan) and then sacrificed by collecting whole blood from the heart. The use of rabbits was approved by the Institutional Review Board of the Shiga University of Medical Science Animal Care and Use Committee (2021-12-5).

## Cleaning of the antiserum with acetone powder

Rat brains (from the frontal lobe to the pons) from *Slc22a23*$^{-/-}$ rats were frozen in liquid nitrogen and then disrupted using a mortar and pestle. Then, the frozen brains were homogenized in 0.9% (w/v) sodium chloride solution using a Dounce homogenizer (7700HOMGI5; IWAKI, Shizuoka, Japan). Acetone powder was prepared from the rat brain homogenate according to the method of Harlow and Lane [18]. After two rounds of phosphate-buffered saline washes, the acetone powder (50 mg) was mixed with antiserum (250 μl). The mixture was then incubated for 30 minutes at room temperature. After centrifugation at 10,000 RCF for 10 minutes, the supernatant was used as the primary antibody for western blotting.

## Western blotting

Brains (from the frontal lobe to the pons) from *Slc22a23*$^{+/+}$, *Slc22a23*$^{+/-}$, and *Slc22a23*$^{-/-}$ rats were frozen in liquid nitrogen and then disrupted using a mortar and pestle. The rat brain powders were dissolved in a urea sample buffer (2.3 M urea, 1.5% SDS, 15 mM Tris pH 6.8, 100 mM DTT, and protease inhibitors (1 mM ABSF, 0.8 μM aprotinin, 15 μM E-64, 20 μM leupeptin, 50 μM bestatin, and 10 μM pepstatin A (Nacalai tesque))) to prepare 100 μg/μl brain samples. The samples were sonicated for 10 seconds using an ultrasonic homogenizer (Branson Ultrasonics, Brookfield, CT, US) and then incubated at 42°C for 30 minutes. After centrifugation at 10,000 RCF for 2 minutes, the samples were loaded onto a sodium dodecyl-sulfate polyacrylamide gel electrophoresis gel. After a semi-dry transfer, the polyvinylidene difluoride (PVDF) membrane (Merk, Darmstadt, Germany) was incubated with an antiserum against SLC22A23 (1:1000). After washing off the primary antibody, the membrane was probed with horseradish peroxidase (HRP)-conjugated anti-rabbit IgG (1:15000) (Cytiva Cat# NA934, RRID:AB_772206). After washing off the secondary antibody, the membrane was immersed in Chemi-Lumi One Super (Nacalai tesque). The signals on the membrane were detected using a chemiluminescence imaging system (FUSION SOLO.6S.EDGE; Viber-Lourmat, Collégien, France). To detect different proteins using the same membrane, the antibodies on the PVDF membrane were stripped using a stripping buffer (2% SDS, 62.5 mM Tris pH 5.5, 100 mM 2-mercaptoethanol) at 60°C for 30 minutes. Then, the membrane was probed with mouse monoclonal antibodies, including anti-CNPase (1:500) (clone 11-5B; Millipore Cat# MAB326, RRID:AB_2082608), anti-tubulin beta III isoform

C-terminus (1:2000) (clone TU20; Millipore Cat# MAB1637, RRID:AB_2210524), or anti-GFAP (1:2000) (clone GA5; Millipore Cat# MAB360, RRID:AB_11212597). After washing off the primary antibodies, the membrane was incubated with HRP-conjugated anti-mouse IgG (1:15000) (Cytiva Cat# NA931, RRID:AB_772210). After washing off the secondary antibody, the membrane was immersed in Chemi-Lumi One L (Nacalai tesque). The signals on the membrane were detected using a chemiluminescence imaging system (FUSION SOLO.6S.EDGE; Viber-Lourmat, Collégien, France) and saved in TIF format. The TIF files were opened in ImageJ (version 1.54g; NIH, US) and SLC22A23 protein band intensities were measured using the Analyze function in ImageJ. To compare the relative band intensities of the SLC22A23 protein among genotypes, the band intensities of the SLC22A23 protein were compensated by the band intensities of the loading controls (anti-Tubulin betaIII, anti-GFAP or anti-CNPase).

## Rat husbandry

Rats were housed under a 12-hour light/dark cycle. The lights were switched on at 8:00 am. $Slc22a23^{+/-}$ rats were bred to obtain trios of $Slc22a23^{+/+}$, $Slc22a23^{+/-}$, and $Slc22a23^{-/-}$ littermates. The dams were fed CE-2 chow (4.77% crude fat per weight) (CLEA Japan, Tokyo, Japan) *ad libitum*. The offspring were weaned at P23. The littermate trios were housed in the same cage and fed CE-2 as the normal chow. In the 2022 winter-1 experiment, after the completion of the experiments at 8 weeks of age, the rats were fed HFD-60 (35% crude fat per weight) (Oriental Yeast, Tokyo, Japan) as a high-fat diet from P70 until the day of sampling (16 weeks of age). In the 2022 winter-2 experiment, each rat was housed in a single cage with a wire mesh from P33 to P63 to measure food and water intake. To measure food intake, powdered food (AIN-93G containing milk casein (crude fat 7% per weight) (Oriental Yeast)) was placed in a container (Roden cafe; Oriental Yeast). A normal water bottle in the animal facility was used to measure the water intake of the rats.

## Behavioral tests

The use of rats in behavioral studies was approved by the Institutional Review Board of the Shiga University of Medical Science Animal Care and Use Committee (2020-9-6, 2021-4-14). Surplus rats were sacrificed by an overdose of carbon dioxide using the euthanasia equipment provided in the animal facility. If the animals were found to be injured during rearing, they were also sacrificed by an overdose of carbon dioxide. Before dissection, the rats were anesthetized by 4% isoflurane inhalation (VIATRIS, Tokyo, Japan) and then sacrificed by collecting whole blood from the heart. **S1 Fig** provides a schematic representation of the behavioral tests. Gender and litter-matched groups of $Slc22a23^{+/+}$, $Slc22a23^{+/-}$, and $Slc22a23^{-/-}$ rats were used and compared in the behavioral tests. Four types of behavioral tests were performed in this study. (1) Open field test. The open field test was conducted in a gray polyvinyl chloride circular chamber (100 cm diameter, 45 cm height). The brightness of the surface of the open field was set to less than 8 lux using a digital lux meter (YF-881D; INFURIDER, Zhuhai, China). Initially, a test rat was placed in the center of the open field. The movement of the test rat (movements of more than 1 cm) was recorded using a CompACT video tracking system (Muromachi Kikai, Tokyo, Japan) for 10 minutes. After each rat was tested, the open field was sprayed with 10% v/v ethanol solution and wiped with disposable paper towels. (2) Novel object recognition test. The same circular chamber used in the open field test was used in this test. Three types of objects were used in the novel object recognition tests. Object X was a shaded round glass bottle (Corning, NY, US) with an orange cap and was used as a familiar object. Object Y was a transparent square glass bottle (Iwaki) with a blue cap that was filled with 3A 1/8 molecular sieves (Wako) and was used as a novel object at 8 weeks. Object Z was a

transparent round glass bottle (Corning) with a blue cap that was filled with clear marbles (17 mm diameter; Matsuno hobby, Osaka, Japan) and used as a novel object at 12 weeks. On day 1, two identical objects (object X) were placed in the open field. The centers of each object were 25 cm from the wall and 50 cm from each other. Then, a test rat was acclimatized to the field for 10 minutes. On day 2, two objects were placed in the open field. One object was the same as that used on day 1 (object X) as a familiar object. The other object was object Y, which was used at 8 weeks of age, or object Z, which was used at 12 weeks of age. A test rat was initially placed in the center of the field. The movement of the test rat (movements of more than 1 cm) was recorded using a CompACT video tracking system (Muromachi Kikai) for 10 minutes. After each rat was tested, the open field and the objects were sprayed with 10% v/v ethanol solution and wiped with disposable paper towels. (3) Elevated plus-maze test. The apparatus had four arms (50 cm length, 10 cm width, 51.5 cm height). Two of the arms lacked walls (open arms). The other two arms had walls (44 cm height) (closed arms). The brightness of the central square was set at 9 lux using a digital lux meter (YF-881D; INFURIDER). The apparatus was set without shade, even inside the closed arms. A test rat was initially placed in the central square of the elevated plus-maze. The movement of the test rat (movements of more than 1 cm) was recorded using a CompACT video tracking system (Muromachi Kikai) for 5 minutes. The time spent in each arm and the number of entries into each arm were recorded. After each rat was tested, the apparatus was sprayed with 10% v/v ethanol solution and wiped with disposable paper towels. (4) Social interaction test. The social interaction test was conducted in a gray polyvinyl chloride circular chamber (150 cm diameter, 45 cm height). The brightness of the surface of the open field was set to less than 8 lux using a digital lux meter (YF-881D; INFURIDER). Two identical circular cages (20 cm diameter, 20 cm height) made of stainless-steel wire were used to house a male intruder rat. The intruder rats (rat A and rat B) were male Slc:Wistar rats purchased from an animal supplier (Japan SLC). The intruder rats were kept isolated from the test rats until the social interaction test was performed. On day 1, two empty circular cages were placed in the field. The centers of each cage were 30 cm from the wall and 90 cm from each other. On day 2, an intruder rat (rat A) was placed in one of the stainless-steel cages, and then a test rat was placed in the center of the field. The movement of the test rat (movements of more than 1 cm) was recorded using a CompACT video tracking system (Muromachi Kikai) for 10 minutes. The first test on day 3 was carried out as on day 2. For the second test on day 3, the intruder rat was replaced by rat B. After each rat was tested, the field and the stainless-steel cages were sprayed with 10% v/v ethanol solution and wiped with disposable paper towels.

## Physical assessment

Three types of physical assessments were performed in this study. (1) Forelimb traction test. A digital force gauge (MK-380Si; Muromachi Kikai) was placed at the edge of a table (70 cm high). A piece of stainless-steel mesh attached to the instrument was protruding from the table. The test rat was allowed to grasp the mesh. The base of the rat's tail was slowly pulled back horizontally. The forelimb traction measurement was repeated five times with an interval of more than 5 minutes between each test. The maximum forelimb traction (kg) of the five measurements was reported. (2) Treadmill test. The angle of the treadmill (TMW-4; Melquest, Toyama, Japan) was set at an upward slope of 10˚. The treadmill speed was set as follows: 15 cm/second for 5 minutes in the beginning, the speed was then increased from 15 cm/second to 40 cm/second over 15 minutes, and the speed was maintained at 40 cm/second until exhaustion. Before the treadmill test, the rats were trained on the treadmill program for 15 minutes daily for 2 days. The treadmill electrode was set to 1.0 mA at the time of training only. The

electrode was turned off during the test. A rat was considered fatigued if it touched the electrode for more than 5 seconds continuously or if it touched the electrode five times within 30 seconds. (3) Four-limb hanging test (Kondziella's inverted screen test) [19]. A rat was allowed to grasp a stainless-steel grid (13 mm square lattice of 1 mm wide stainless-steel wire) with four legs. Then, the grid was turned over and held 45 cm above the ground. The duration that the rat could hold the grid was measured. The four-limb hanging test was repeated three times with an interval of more than 30 minutes between each test. The maximum duration of the hanging of the three measurements was reported.

## MR imaging

MR images were acquired using a 4.7 Tesla Bruker BioSpec 47/40 USR (Bruker BioSpin GmbH, Ettlingen, Germany) with an insert gradient coil (B-GA12SHP; Bruker BioSpin GmbH) and a radiofrequency coil (1P T20070V3; Bruker BioSpin GmbH). The test rat was anesthetized with a mixture of oxygen and isoflurane (4% isoflurane during induction, then 2%–4% for maintenance). The isoflurane level was adjusted to maintain a respiratory rate of 45–50 breaths/minute throughout MR image acquisition. The program used for MR image acquisition was T2-weighted turbo Rapid Acquisition with Relaxation Enhancement (RARE), with the following parameters: echo spacing = 12 ms, echo time = 36 ms, RARE factor = 8, repetition time = 4800 ms, excitation angle = 90˚, refocusing angle = 180˚, image size = 256 x 256, field of view = 35 x 35 mm, slice thickness (along the rostral/caudal direction) = 0.8 mm, number of slices = 37, scan time = 10 minutes, 14 seconds.

## MR image analysis

Files in the DICOM format exported from the ParaVision 6.0.1 system (Bruker BioSpin GmbH) were converted to NIfTI format, and the voxel size was converted to 137 μm isotropic using the utility function of BAAD (version 4.3.2) [20]. Then, the MR images were aligned and oriented to the Waxholm axis [21, 22] using SPM12 (Functional Imaging Laboratory, UCL Queen Square Institute of Neurology, London, UK). The NIfTI files were randomized and anonymized (by YK). For volume estimation, the rat brain region from the tip of the frontal lobe to the midbrain was exported as a series of JPEG images (slice thickness = 0.5 mm, 36 slices) using the Screen Capture module in 3D Slicer (version 5.6.1; 3D Slicer Community). Using the JPEG images, volume estimations of eight brain regions were performed (independently by YU and JU) using Cavalieri's method with Stereo Investigator (version 10.31; mbf bioscience, Williston, VT, US) following the rat brain atlas [23, 24]. The shape factor used in the volume estimation was set to 12. The shape factor was calculated from the measurement of the total boundary length (B) and the total profile area (A) of the corpus callosum because the corpus callosum is the thinnest structure among our target objects; the shape factor was calculated as $B/\sqrt{A}$. The coefficient of error (m = 1) [25] was less than 0.05 for all measurements. For three-dimensional visualization of the hippocampus, the MR images aligned and oriented to the Waxholm axis were analyzed in 3D Slicer (version 5.6.1; 3D Slicer Community). Following the Waxholm space atlas of the rat brain hippocampal region [26], the hippocampal regions were marked using the Segment Editor module in 3D Slicer (version 5.6.1; 3D Slicer Community). The scalar volume of each hippocampus was calculated using the Segment Statistics module in 3D Slicer (version 5.6.1; 3D Slicer Community).

## Statistical analysis

All figures were plotted using GraphPad Prism 10 (Dotmatics, Boston, MA, US). All statistical analyses were performed using JMP pro 16 (JMP Statistical Discovery, Cary, NC, US), except where specifically noted in the figure legends.

## Results

### Prenatal undernutrition increases *Slc22a23* gene transcripts in the rat brain

We obtained control fetuses from pregnant dams that were fed *ad libitum* with a standard diet, AIN-93G. Fetuses subjected to undernutrition were obtained from dams that were fed 40% of the amount of AIN-93G diet consumed by the control dams between gestational day (GD) 5.5 and GD 10.5. The reason for the amount of food given to the undernutrition group (40%) is that during the Dutch famine from December 1944 to April 1945, daily rations decreased to between 400 and 800 calories, and after June 1945, daily rations increased to over 2000 calories (800/2000 calories = 0.4) [27]. mRNA was isolated from the forebrain of the fetuses at GD 10.5 and subjected to microarray analysis. Although the transcripts of several genes were significantly different between these two groups (**Fig 1A and S1 Table**), the transcripts of two genes, *Slc22a23* (fold change (fc) > 2.03, false discovery rate (fdr) = $9.76 \times 10^{-14}$) and gonadotropin-releasing hormone 1 (Gnrh1; fc > 10.44, fdr = $1.63 \times 10^{-15}$) were highly significantly increased in the undernutrition group (**Fig 1A**). Next, we raised the offspring of these dams (fed *ad libitum* or 40% of the amount of food from GD 5.5 to GD 10.5) until 9 weeks of age. Using real-time PCR analysis of RNA samples from the prefrontal cortex of these offspring, we confirmed that the transcript of the *Slc22a23* gene, but not the *Gnrh1* gene, was significantly increased in the undernutrition group compared with that in the control group (fc > 2.35, p < 0.015) (**Fig 1B**). These results suggest that undernutrition during fetal development increases *Slc22a23* gene expression, and its overexpression is maintained throughout life (or at least into adulthood). Although we hypothesized that *Slc22a23* is the key gene underlying the undernutrition-induced DOHaD phenotype, the substrate of the SLC22A23 transporter has not been identified and the biological role of the *Slc22a23* gene *in vivo* has not been studied in detail.

### Generation of *Slc22a23* knockout rats

We generated a floxed strain, called W-floxed-*Slc22a23*, in which two loxP sequences were inserted into the introns flanking exon 4 of the *Slc22a23* gene (**Fig 2A**). LoxP-inserted rats were selected by PCR analysis of their tail tips using the F1 and R1 primers (**Fig 2B**). Sequencing analysis confirmed the "clean" targeted insertion of two loxP sequences as designed in the W-floxed-*Slc22a23* strain. We then obtained $Slc22a23^{+/-}$ rats by crossing the W-floxed-*Slc22a23* rats with Cre driver rats (W-Tg(CAG-cre)81Jmsk (NBRP-Rat No. 0283)). Skipping exon 4 of the *Slc22a23* gene via loxP site-specific recombination results in a frameshift and production of a truncated form of the SLC22A23 protein (**Fig 3A**). By breeding $Slc22a23^{+/-}$ rats, we obtained $Slc22a23^{+/+}$, $Slc22a23^{+/-}$, and $Slc22a23^{-/-}$ littermates in a Mendelian ratio (**Fig 3B and 3C**). Using an antiserum against the C-terminal region of the SLC22A23 protein (**Fig 3A**), we confirmed that the full length of the SLC22A23 protein is not detectable in the brain extract from $Slc22a23^{-/-}$ rats (**Fig 3D**).

### $Slc22a23^{-/-}$ rats have reduced body weight

After weaning, the $Slc22a23^{+/+}$, $Slc22a23^{+/-}$, and $Slc22a23^{-/-}$ littermates were reared in the same cage to ensure that they were raised in the same environment. Both male and female $Slc22a23^{-/-}$ rats showed significantly lower body weight than the $Slc22a23^{+/+}$ rats (**Fig 4**). In contrast, except at postnatal day (P)27 and P30 of the 2022 winter-1 experiment, the $Slc22a23^{+/-}$ rats had no significant difference in weight compared with $Slc22a23^{+/+}$ rats (**Fig 4B**). We examined the weight gain, food intake, and water intake of $Slc22a23^{+/+}$, $Slc22a23^{+/-}$, and $Slc22a23^{-/-}$ rats from P33 to P63 by housing them individually in cages (**S2 Fig**). We found no statistically significant difference in weight gain, food intake, or water intake among these genotypes (**S2 Fig**).

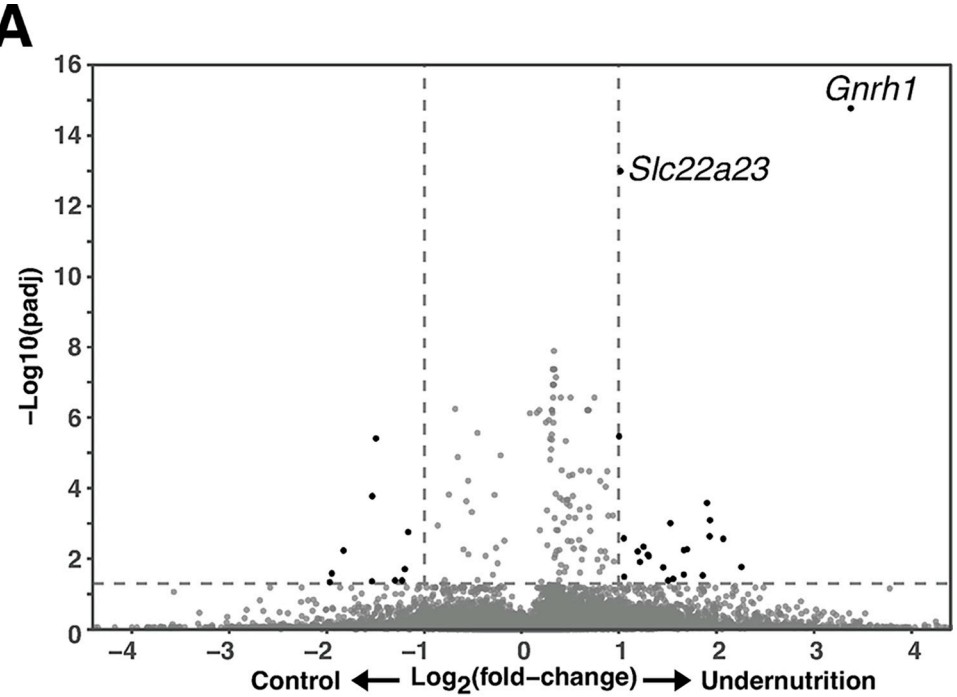

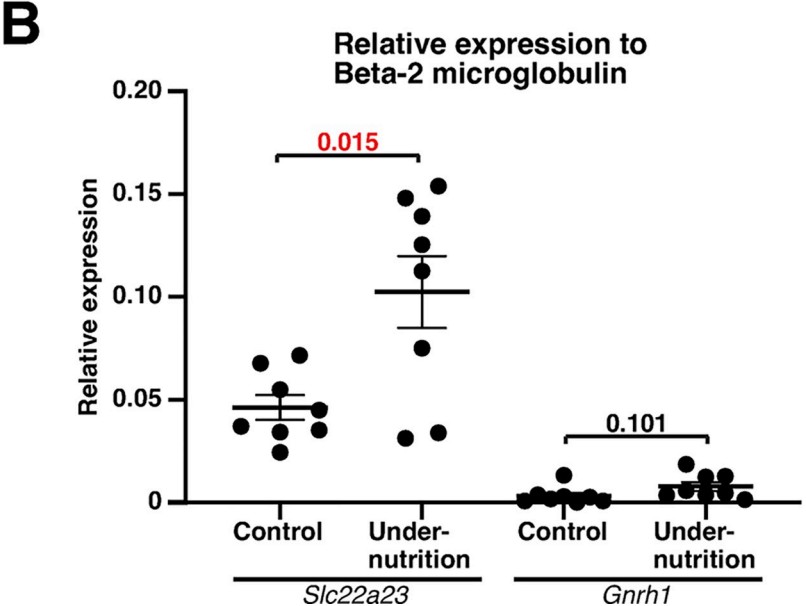

**Fig 1. Increased *Slc22a23* gene transcripts in the rat brain caused by prenatal undernutrition.** (A) RNA array analysis of mRNA isolated from the forebrain at gestational day (GD) 10.5 (n = 3 in each group). The fetuses of the undernutrition group were isolated from pregnant dams that were fed 40% of the amount of food consumed by *ad libitum*-fed control dams from GD 5.5 to GD 10.5. The padj value represents the adjusted p-value calculated by the sumz function of the metap package in R. (B) Real-time PCR analysis of mRNA in the prefrontal cortex of undernutrition and control rats at 9 weeks of age (n = 8 in each group). The undernutrition rats experienced undernutritional stress during fetal development from GD 5.5 to GD 10.5 as described in (A). The error bars show the mean ± standard error of the mean (SEM). The p-values indicate two-tailed *t*-tests.

**A**

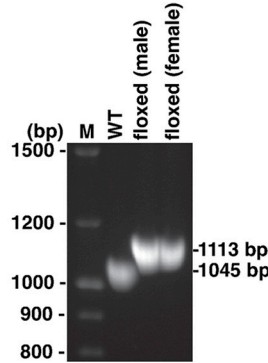

F1: AACAGGTTCTTGTCAACAGGAGGGACCATT
AATGGATCTTGGCCAGCAGGCTTCAGTATG
TATGGTACAGGGGCATTCCTGCTAGGTTAT
AGTATGCCCCAAGTGGCTTTGCTGTTTGTT
TTGTGCAGTATGGTCGGTCTGTTTTGTTTT
CGAAGTGGAGATGAACTTAGAGTCTTCACA
TAGCTGGTGTAAGACAAGGAGCAAAGGATG
GTTGTAGGGGCAGGAATTGAATAGAGAGAC
CTTGCAGAGGCTGAGAGAGCCCCAGTGCAC
AACCTAGGTTGGAAGGAAAGGCGATGCCAT
GGGCACCAAGGGGACATGCTGGGCAGCAAG
CTCTGTGATGCCAAAGAGCAAAGGAAAGAG
GATAATGAGAGATTCAAATTCCGTCTCCAG
TAGCAGCAGCAACCCATCTTAGCTCCTGAG
ATAACTTCGTATAATGTATGCTATACGAAG
TTATCTAAGCACTTCTAAATGTACCATACT
TCCCACTAGCTTGTTCCATGACCTTGAGCA
ACAACCTTACTCTTTTTCCCATCAGGCCTC
AGTTCCCCCATGTATCCAGTGACATCTCAG
GGCTCCTCAGGCATCAGACAAGCAAGTGAC
ACTCAATCTACCTTATTATTTCTGTTTTCT
AGGAATAGAGCTGTGTCCCCCAGGAAAACG
GTTCATAATCACCATGGTGGCGAGCTTCGT
GGCCATGGCAGGCCAGTTCCTCATGCCCGG
GCTGGCTGCGCTGTGCCGGGATTGGCAGGT
TTTGCAGGCCCTCATCATCTGTCCCTTCCT
GCTCATGCTGCTCTACTGGTCGTGAGTATT
CTTTGTGACGTACCACATCATACATGCCTG
GCTCGCTGCAGGCCCTGCAGGCTCTGACCT
CCATTGTGTTAACATGTTTTTCCTGACAAG
GTCTCACCTCAATAACTTCGTATAATGTAT
GCTATACGAAGTTATGATAGGTCTTTCTCA
ACACATTCACTCACAGATACTCACATTCTC
TGTCTCTGTCTCTGTCTGTCTCTCCCTCCC
TCCCAGCTCCCTGTGTGTGTGTGTGTGTGT
GTGTGTGTGTGTGTGTGTGTGTGTGAACTG
TTTACATGGTTCA<u>TAATGTGAAATGAGGCC
CTC</u> R1

**B**

**Fig 2. Generation of floxed *Slc22a23* rats.** (A) The target sites of the loxP sequences used to generate floxed *Slc22a23* rats. The loxP sequences (shown in red) were inserted into the introns (shown in black) flanking exon 4 (shown in blue) of the *Slc22a23* gene. The underlined F1 and R1 sequences were used as primers for PCR analysis. (B) PCR analysis of floxed *Slc22a23* rats. Tail genomic DNA from floxed rats and a Jcl:Wistar rat (wild-type control, WT) was amplified using the F1 and R1 primers. M represents a 100-bp DNA ladder marker. The raw image of Fig 2B can be found in the S1 Raw images.

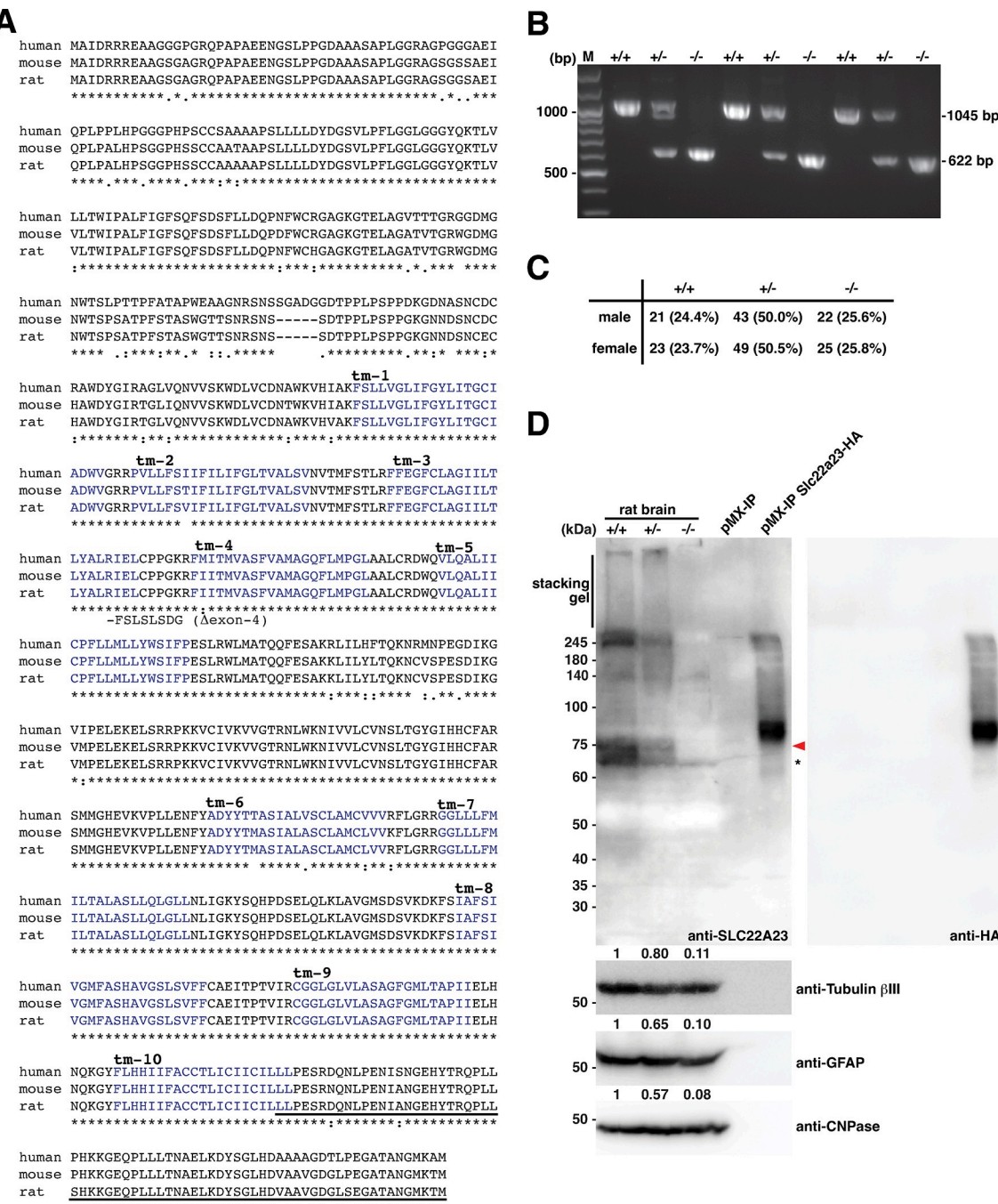

**Fig 3. Generation of *Slc22a23* knockout rats.** (A) Alignment of SLC22A23 amino acid residues. The amino acid residues of human SLC22A23 (A1A5C7), mouse SLC22A23 (A1A5C7), and rat SLC22A23 (Q9QZG1) were aligned using CLUSTALW (https://www.genome.jp/tools-bin/clustalw). tm-1 –tm-10 represents the predicted transmembrane domains in UniProt (https://www.uniprot.org/). The C-terminal region used for immunization to raise antiserum against rat SLC22A23 is underlined. (B) Tail PCR genotyping of litters resulting from the breeding of *Slc22a23*<sup>+/−</sup> pairs. The raw image of Fig 3B can be found in the S1 Raw images. (C) The ratio of offspring genotypes of *Slc22a23*<sup>+/−</sup> parents. (D) Western blot of the rat brain lysate from *Slc22a23*<sup>+/+</sup>, *Slc22a23*<sup>+/−</sup>, and *Slc22a23*<sup>−/−</sup> rats. Lysate from C2C12 cells transfected with pMX-IP plasmid was used as a negative control. Lysate from C2C12 cells transfected with pMX-IP Slc22a23-HA plasmid was used as a positive control. SLC22A23 protein band intensities, indicated by the red arrowhead, were measured in ImageJ (version 1.54g; NIH, US). To compare the relative band intensities of the SLC22A23 protein among genotypes, the band intensities of the SLC22A23 protein were compensated by the band intensities of the loading controls (anti-Tubulin betaIII, anti-GFAP or anti-CNPase). The values of relative band intensities of the SLC22A23 protein compared to +/+ are shown. The raw image of Fig 3D can be found in the S1 Raw images.

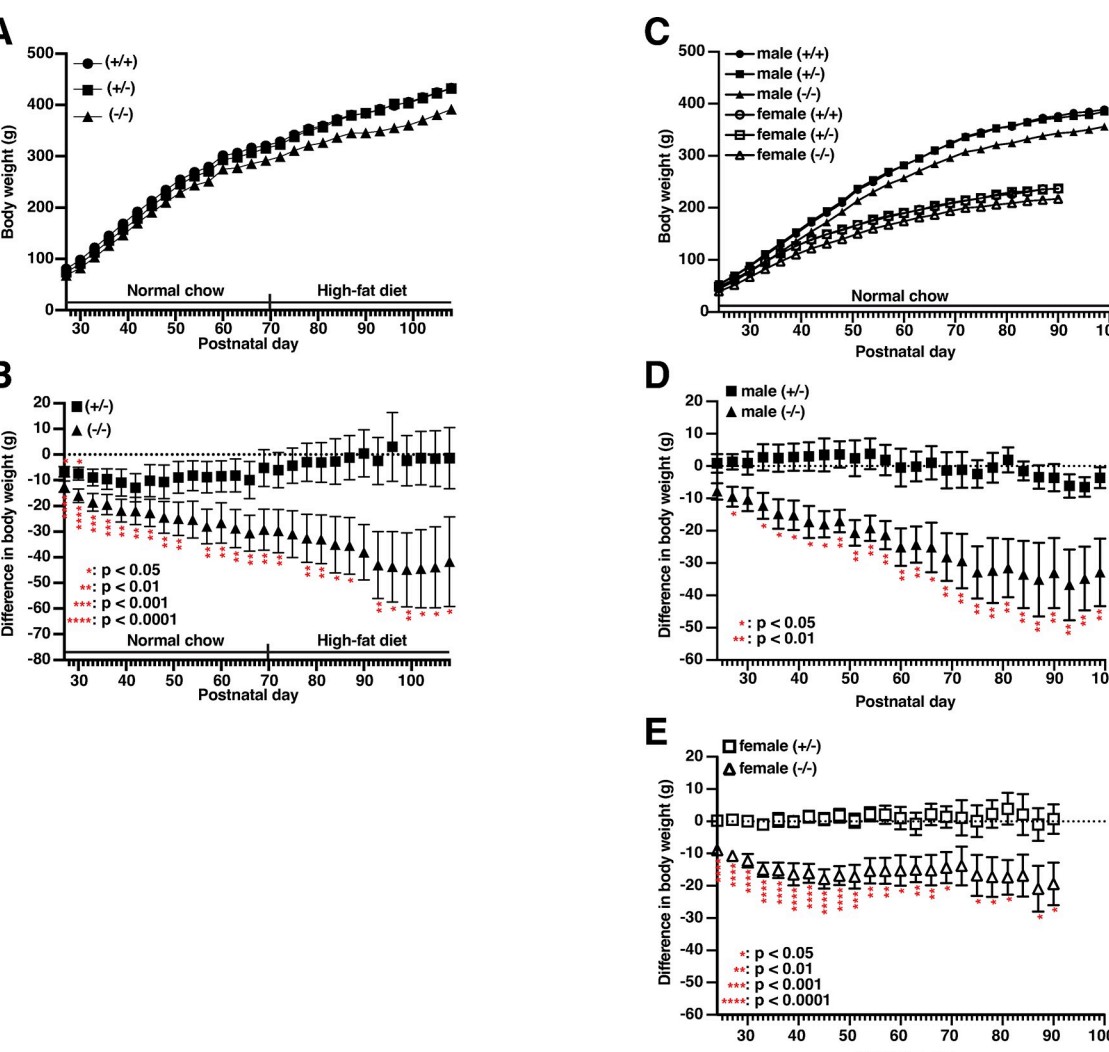

**Fig 4. Reduced body weight in *Slc22a23* knockout rats.** (A) Mean body weight of male *Slc22a23*[+/+], *Slc22a23*[+/−], and *Slc22a23*[−/−] rats in the 2022 winter-1 experiment. The body weights were measured every 3 days (n = 7 in each group). Normal chow (CE-2) was given before conception until postnatal day (P)70, and then a high-fat diet (HFD-60) was given from P70 until sacrifice. (B) Difference in body weight compared with corresponding +/+ littermates. (C) Mean body weight of male and female *Slc22a23*[+/+], *Slc22a23*[+/−], and *Slc22a23*[−/−] rats in the 2023 summer experiment. The body weights were measured every 3 days (n = 5 in male groups, n = 7 in female groups). Normal chow (CE-2) was given throughout the experiment. (D) Difference in body weight (males) compared with corresponding +/+ littermates in the 2023 summer experiment. (E) Difference in body weight (females) compared with corresponding +/+ littermates in the 2023 summer experiment. The error bars show the mean ± SEM. The p-values were calculated using Dunnett's multiple comparisons with control +/+ rats.

### *Slc22a23*[−/−] rats show increased spontaneous exploratory movements

In the open field test, male *Slc22a23*[−/−] rats showed a statistically significant increase in the total distance travelled compared with control +/+ rats at both 8 weeks (**Fig 5A1**: fc = 1.34, p = 0.011; **S3A Fig**: fc = 1.32, p = 0.042) and 12 weeks of age (**Fig 5A1**: fc = 1.42, p = 0.001; **S3A Fig**: fc = 1.36, p = 0.025). Consistent with this observation, the frequency of the crossing times of male *Slc22a23*[−/−] rats was significantly increased at both 8 weeks (**Fig 5A2**: fc = 1.30, p = 0.025; **S3B Fig**: fc = 1.32, p = 0.032) and 12 weeks of age (**Fig 5A2**: fc = 1.40, p = 0.001; **S3B Fig**: fc = 1.33, p = 0.016). In contrast, female *Slc22a23*[−/−] rats showed no statistically significant difference compared with control +/+ rats in the open field test (**S3 Fig**). Corresponding to the

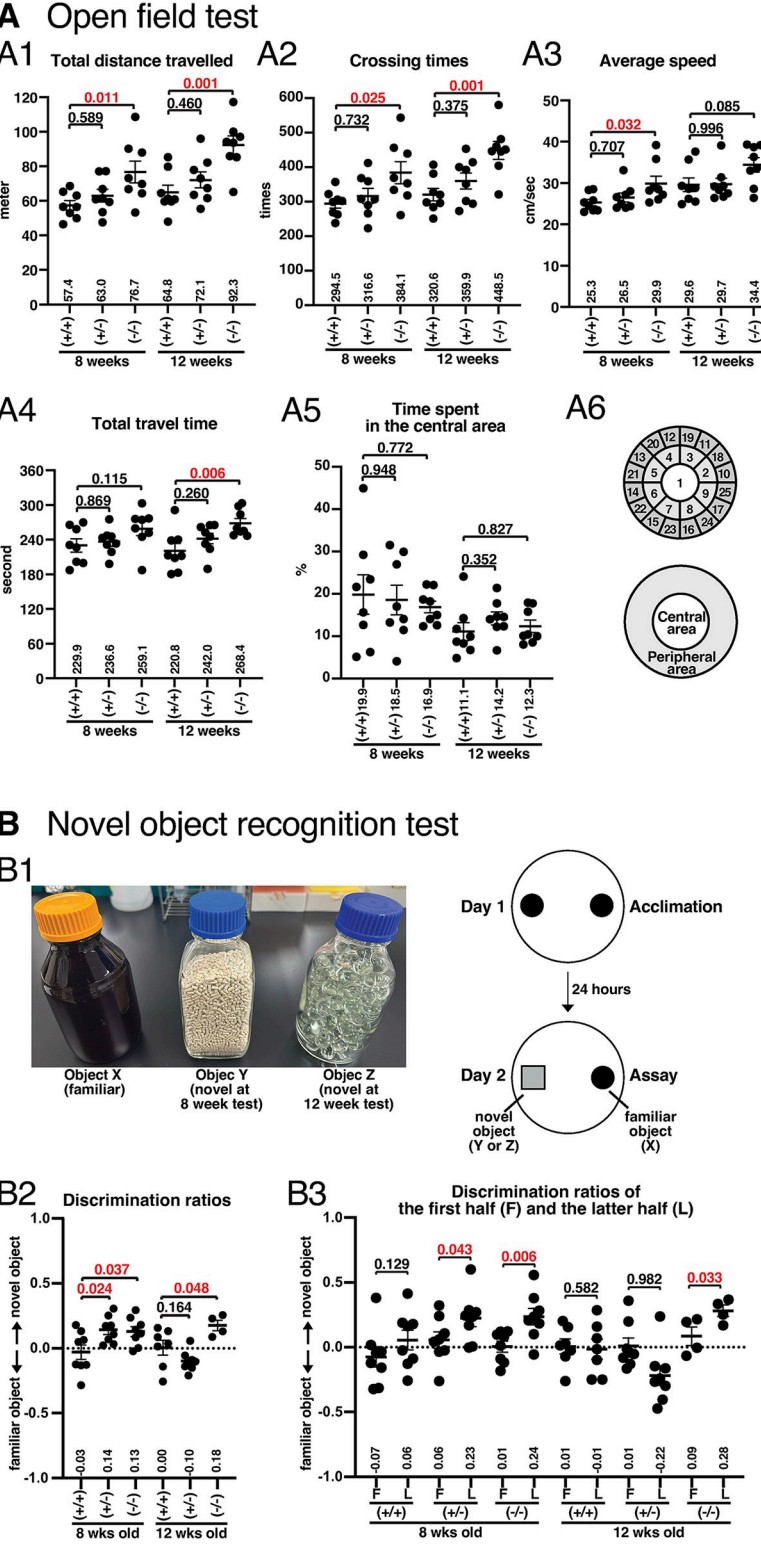

**Fig 5. Increased total distance travelled and altered novel object recognition in *Slc22a23* knockout rats.** (A) Open field test. (A1) Total distance travelled in the open field. A test rat was placed in the center of an open field (100 cm diameter, 45 cm height), and movements of more than 1 cm were recorded for 10 minutes. The rats were tested at 8 weeks of age, and the same rats were tested 4 weeks later at 12 weeks of age. Male *Slc22a23*^+/+^, *Slc22a23*^+/−^, and *Slc22a23*^−/−^ rats (n = 8 in each group) in the 2022 winter-1 experiment are shown. (A2) Crossing times. The open field

was divided into 25 subregions as shown in A6. The number of times each border was crossed (2 cm wide) was counted. (A3) Average travelling speed. The average speed at which a test rat moved (more than 1 cm) was recorded. (A4) Total travel time in seconds. The total duration that the rat performed movements of more than 1 cm was recorded. (A5) Percentage of time spent in the central area. The open field was divided into a central and a peripheral area as shown in A6. The central and peripheral areas are the same size. The percentage of duration spent in the central area was calculated. The error bars show the mean ± SEM. The p-values were calculated using Dunnett's multiple comparisons with control +/+ rats. The factorial ANOVA of the data in Fig 5A can be found in Panel A of S2 Table. (B) Novel object recognition test. (B1) The objects and scheme in the novel object recognition test. On day 1, two identical objects (object X) were placed in the open field (100 cm diameter, 45 cm height), and then a test rat was acclimatized for 10 minutes. On day 2, a familiar object (object X) and a novel object (object Y at 8 weeks of age; object Z at 12 weeks of age) were placed in the open field. After placing a test rat in the center of the open field, the rat's movement was recorded for 10 minutes. The duration of the approach to each object was assessed when the rat's head was within 7 cm of the wall of each bottle. (B2) Discrimination ratios in the novel object recognition test. The error bars show the mean ± SEM. The p-values were calculated using Dunnett's multiple comparisons with control +/+ rats. In the test at 12 weeks of age, the results of one *Slc22a23*^(+/+) rat and four *Slc22a23*^(−/−) rats were not obtained because they climbed onto the lid of the object. The factorial ANOVA of the data in Fig 5B2 can be found in Panel C of S2 Table. (B3) Discrimination ratios in the first and latter half of the assay. F indicates the first half (0–5 minutes). L indicates the latter half (5–10 minutes). The error bars show the mean ± SEM. The p-values show the results of a one-tailed *t*-test. It is assumed that the discrimination ratio for a novel object will not be higher in the latter half than in the first half.

increase in the total distance travelled in male *Slc22a23*^(−/−) rats, both the average speed of movement (**Fig 5A3**: fc = 1.18, p = 0.032 at 8 weeks of age) and total travel time (**Fig 5A4**: fc = 1.22, p = 0.006 at 12 weeks of age; **S3D Fig**: fc = 1.27, p = 0.022 at 12 weeks of age) were significantly increased in male *Slc22a23*^(−/−) rats. Neither the time spent in the central area (**Fig 5A5 and S3E Fig**) nor the elevated plus maze results (**S4 Fig**) were significantly different from those of control +/+ rats. These results suggest that the absence of the *Slc22a23* gene has no significant effect on the anxiety/anti-anxiety phenotype.

## Recognition of novel objects is prolonged in *Slc22a23*^(−/−) rats

In the novel object recognition test, the approach to a novel object (object Y at 8 weeks of age; object Z at 12 weeks of age) was measured as the duration of access when the head of a test rat was within 7 cm of the wall of the object (**Fig 5B1**). The discrimination ratio (d2) was calculated using the following formula: d2 = [time (novel object)–time (familiar object)]/time (novel object + familiar object) [28]. The discrimination ratios were significantly increased in *Slc22a23*^(−/−) rats compared with those of control +/+ rats (**Fig 5B2**: p = 0.037 at 8 weeks; p = 0.048 at 12 weeks). To examine the findings in detail, the discrimination ratios of the first half of the assay (5 minutes) were compared with those of the latter half. The absence of the *Slc22a23* gene induced a statistically significant difference in access to the novel object in the latter half of the assay compared with that in the first half (**Fig 5B3**: p = 0.006 at 8 weeks; p = 0.033 at 12 weeks), suggesting a reduced efficiency of the object recognition in the deficiency of the *Slc22a23* gene.

## *Slc22a23*^(−/−) rats show a minor difference in social interaction

The social interactions of *Slc22a23*^(+/+), *Slc22a23*^(+/−), and *Slc22a23*^(−/−) rats with intruder rats were examined by applying the method described for the novel object recognition test using intruder rats (rat A and rat B) (**Fig 6A**). As in the novel object recognition test, the approach to a target object was measured as the duration of access when the head of a test rat was within 7 cm of the wall of the object. The discrimination ratio (d2) was calculated using the following formula: d2 = [time (quest cage)–time (empty cage)]/time (quest cage + empty cage). Although no statistically significant difference was found in the discrimination ratios between rat A on day 2 and day 3 (**Fig 6B**), we found a statistically significant difference in the discrimination ratios of *Slc22a23*^(−/−) rats between rat A and rat B on day 3 (**Fig 6C**: p = 0.028). However, the

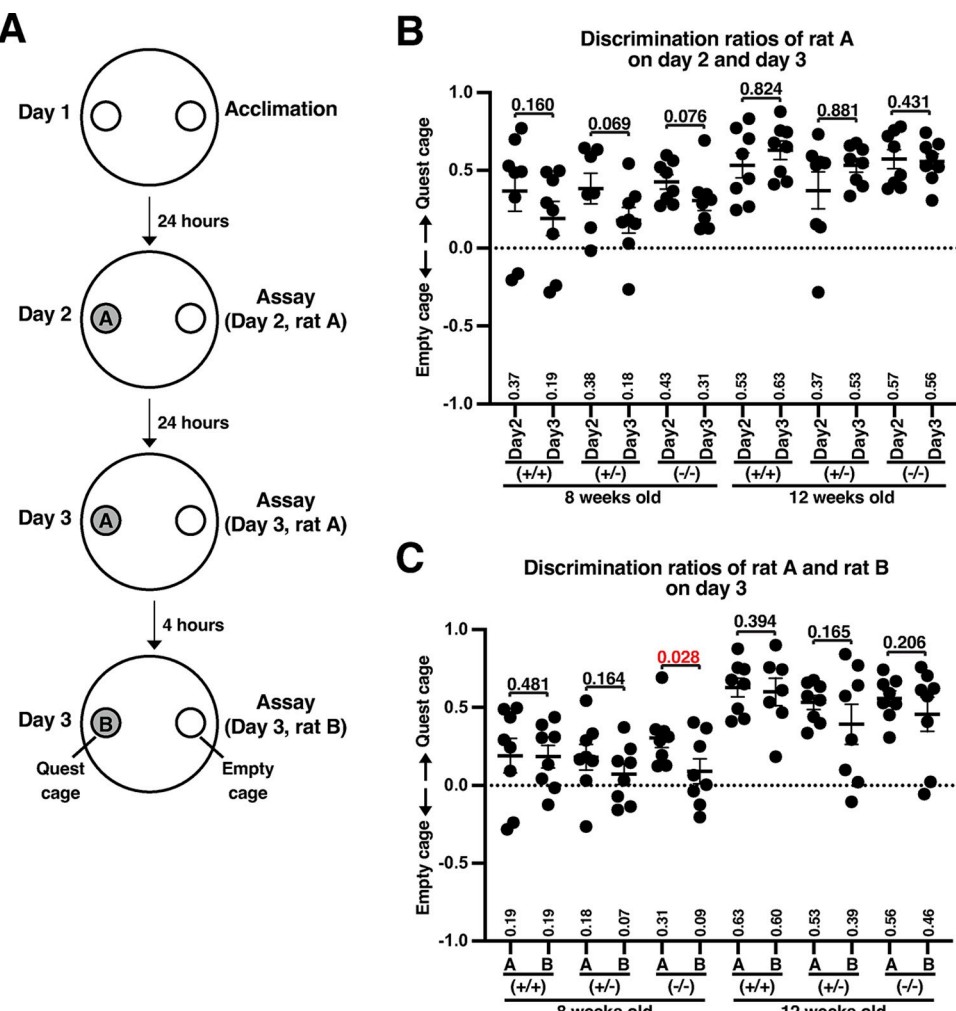

**Fig 6. *Slc22a23* knockout rats showed a small but significant difference in the social interaction test.** (A) The scheme of the social interaction test. On day 1, two identical empty circular cages (20 cm diameter, 20 cm height) were placed in the circular chamber (150 cm diameter, 45 cm height). A test rat was acclimatized in the field for 10 minutes. On day 2, an intruder rat (rat A) was placed in one of the circular cages, and then a test rat was introduced in the center of the field. On day 3, the first test was carried out as on day 2. The second test on day 3 was carried out by replacing rat A with rat B. The movement of test rats was recorded using a CompACT video tracking system (Muromachi Kikai) for 10 minutes. When the head of a test rat was within 7 cm of the wall of the cage, the rat was considered to have accessed the cage. (B) Comparisons of the discrimination ratios of rat A between day 2 and day 3. The factorial ANOVA of the data in Fig 6B can be found in Panel D of S2 Table. (C) Comparisons of the discrimination ratios between rat A and rat B on day 3. The discrimination ratios in the first half of the test (5 minutes) are shown. The error bars show the mean ± SEM. The p-values show the result of a one-tailed t-test. It is assumed that the discrimination ratio of the quest cage in the latter assay will not be higher than the ratio in the former assay. The factorial ANOVA of the data in Fig 6C can be found in Panel E of S2 Table.

overall trends of the discrimination ratios at 8 weeks of age were reduced by repeated measurements in *Slc22a23*$^{+/+}$, *Slc22a23*$^{+/-}$, and *Slc22a23*$^{-/-}$ rats (**Fig 6B and 6C**). We concluded that the difference in social interaction in the absence of the *Slc22a23* gene is minor.

## *Slc22a23*$^{-/-}$ rats show increased endurance/stamina

Because *Slc22a23*-deficient rats have a lean phenotype (**Fig 4**), we examined slow and fast muscle functions. The maximum traction force was measured using a digital force gauge.

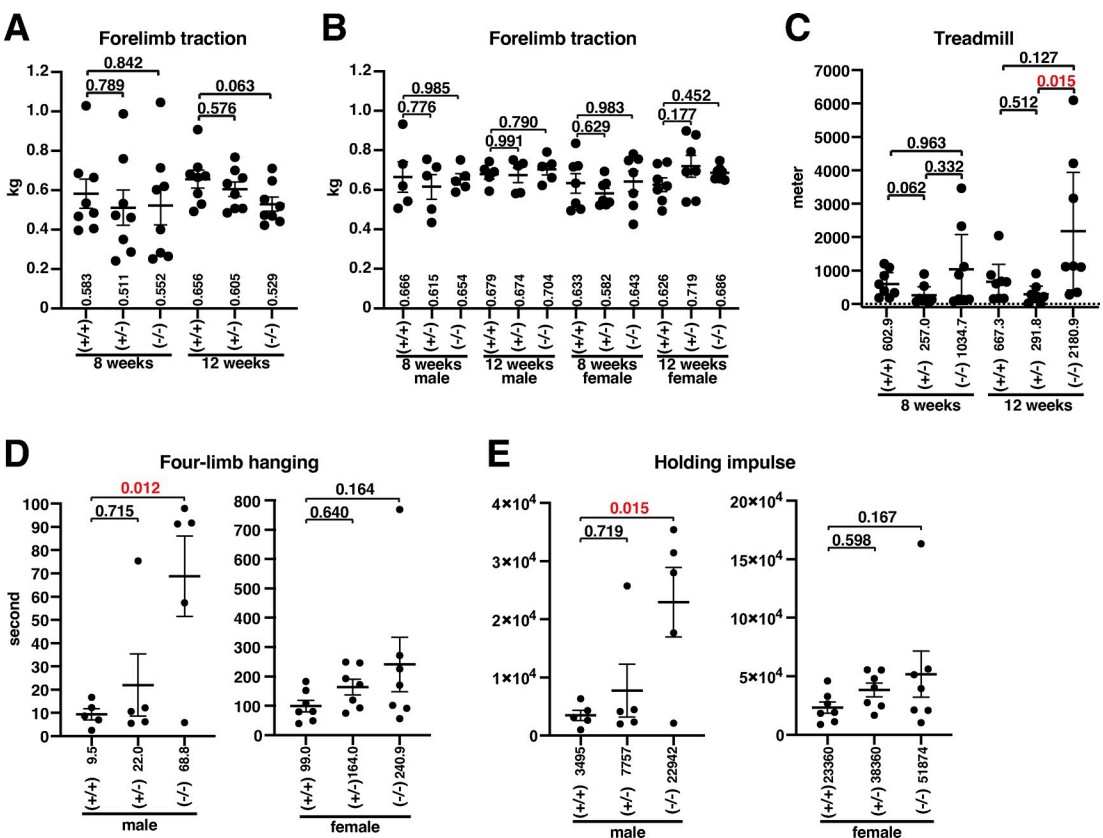

**Fig 7. *Slc22a23* knockout rats exhibited improved performance in slow muscle function tests.** (A) Maximum forelimb traction force. A digital force gauge (MK-380Si; Muromachi Kikai) was used to measure the maximum forelimb traction force. After a test rat grasped the grid of the device with its forelimb, the rat's tail was slowly pulled back horizontally. The maximum forelimb traction force in five trials is plotted. The rats were tested at 8 weeks of age, and the same rats were tested 4 weeks later at 12 weeks of age. Male *Slc22a23*⁺/⁺, *Slc22a23*⁺/⁻, and *Slc22a23*⁻/⁻ rats (n = 8 in each group) in the 2022 winter-1 experiment are shown. (B) Maximum forelimb traction force in the 2023 summer experiment (n = 5 in male groups, n = 7 in female groups). (C) Treadmill test. The running endurance of rats was measured using a treadmill (TMW-4; Melquest). After 2 days of daily training for 15 minutes, the rats were tested to determine their running endurance until exhaustion. A rat was considered exhausted if it touched the electrode continuously for more than 5 seconds or if it touched the electrode five times within 30 seconds. The electrode was turned on only during training and was turned off during the test. The error bars show the mean ± SEM. Because the running distance showed a skewed distribution, the p-values were calculated using nonparametric multiple comparisons with the Steel-Dwass method, all pairs. (D) Four-limb hanging test (Kondziella's inverted screen test). After a test rat grasped a stainless-steel grid (13 mm square lattice, 1 mm wide, stainless-steel wire) with four legs, the grid was turned over and held 45 cm above the ground. The duration the rat could hold the grid was measured. The maximum duration of hanging in three trials at 12 weeks of age in the 2023 summer experiment (n = 5 in male groups, n = 7 in female groups) is shown. (E) Holding impulse. The holding impulse (body mass (grams) x hanging time (seconds)) was calculated to compensate for the difference in body weight. The error bars show the mean ± SEM. Except for C, the p-values were calculated using Dunnett's multiple comparisons with control +/+ rats. The factorial ANOVA of the data in Fig 7 can be found in Panel A-D of S3 Table.

*Slc22a23*⁻/⁻ rats showed no statistically significant difference in the maximum traction force compared with control +/+ rats (**Fig 7A and 7B**), suggesting that the *Slc22a23* gene has a negligible effect on fast muscle function. In contrast, slow muscle function tests showed statistically significant differences between *Slc22a23*-deficient and *Slc22a23*-proficient rats. The treadmill test showed a statistically significant increase in the running distance of male *Slc22a23*⁻/⁻ rats compared with *that of Slc22a23*-proficient rats (**Fig 7C**: fc = 7.47, p = 0.015). Because some *Slc22a23*⁻/⁻ rats could run on the treadmill for more than 2 hours (like a full marathon), we decided to use a shorter method to test slow muscle function in our second experiment. The four-limb hanging test showed a statistically significant increase in the duration of hanging

under the grid in male *Slc22a23*$^{-/-}$ rats compared with that in control +/+ rats (**Fig 7D**: fc = 7.24, p = 0.012), but no significant difference was observed in females (**Fig 7D**). The holding impulse, which is calculated by body weight (grams) × duration of hanging (seconds) [19] to correct the negative effect of body weight on the hanging time, showed a statistically significant increase in male *Slc22a23*$^{-/-}$ rats compared with that in control +/+ rats (**Fig 7E**: fc = 6.56, p = 0.015). Although the slow muscle function tests suggest that male *Slc22a23*$^{-/-}$ rats simply have better slow muscle function than *Slc22a23*-proficient rats, there are other possibilities, such as that the male *Slc22a23*$^{-/-}$ rats have a much greater sense of not giving up. We therefore need to perform other behavioral tests such as a tail-suspension test in the future.

## Brain weight is reduced in *Slc22a23*$^{-/-}$ rats

Immediately before organ sampling, the body length of rats was measured from nose to anus under anesthesia. The body length of *Slc22a23*$^{-/-}$ rats was slightly shorter than that of the other groups (**Fig 8A**). An anthropometrical body mass index (BMI) for rats has been proposed as follows: BMI = body weight (grams)/nose-to-anus length$^2$ (cm$^2$) [29]. Although the anthropometrical BMI was not significantly different from that of control +/+ rats in the population (**Fig 8B**), the *Slc22a23*$^{-/-}$ rats showed a statistically significant reduction in their BMI compared with corresponding +/+ littermates (**Fig 8C**). We then performed organ sampling. The brain weight was significantly reduced in *Slc22a23*$^{-/-}$ rats compared with that of control +/+ rats (**Fig 8D1**), although the weights of the liver, kidney, heart, and spleen in *Slc22a23*$^{-/-}$ rats were not significantly different from those of control +/+ rats (**Fig 8D2–8D5**, **S5B and S5C Fig**). The reduction of brain weight was consistent even when the rats were fed normal chow or a high-fat diet (**S5A Fig**). Next, the brains isolated in Fig 8D1 were further divided into four regions: (1) cerebral hemisphere and diencephalon, (2) midbrain, (3) pons, and (4) cerebellum. The weight of the cerebral hemisphere and diencephalon was significantly reduced in *Slc22a23*$^{-/-}$ rats compared with that in control +/+ rats (**Fig 8E1**: 62 mg, p = 0.008), but no significant difference was found for the other regions (**Fig 8E2–8E4**).

## *Slc22a23*$^{-/-}$ rats have reduced hippocampal volume

Magnetic resonance (MR) images were obtained 1 week before organ sampling (**Fig 8D**). The volumes of eight brain regions (hippocampus, cortex, striatum, corpus callosum, ventricles, thalamus, hypothalamus, and periaqueductal gray) were estimated using the Cavalieri estimator function of Stereo Investigator. The Cavalieri's estimated hippocampus volume was significantly reduced in *Slc22a23*$^{-/-}$ rats compared with that in control +/+ rats (**Fig 8F1**: 10.16 mm$^3$, p = 0.032), although the volumes of other regions were not significantly different (**Fig 8F2–8F8**). According to the Waxholm space atlas of the rat brain hippocampal region [26], three-dimensional surface models of the hippocampus were constructed (**Fig 9A**). The scalar hippocampal volume was significantly reduced in *Slc22a23*$^{-/-}$ rats compared with that in control +/+ rats (**Fig 9B**: 10.03 mm$^3$, p = 0.009).

## Discussion

### Phenotype of *Slc22a23* knockout rats

In this study, we generated *Slc22a23* knockout rats (**Fig 3**) and analyzed their phenotype to elucidate the function of the SLC22A23 transporter *in vivo*. The *Slc22a23* knockout rats showed a lean phenotype (**Fig 4**), suggesting that the SLC22A23 transporter is involved in the uptake of nutrients required for weight gain, although the putative nutrients have not been identified. Throughout most of our evolutionary history, the nutritional environment has been

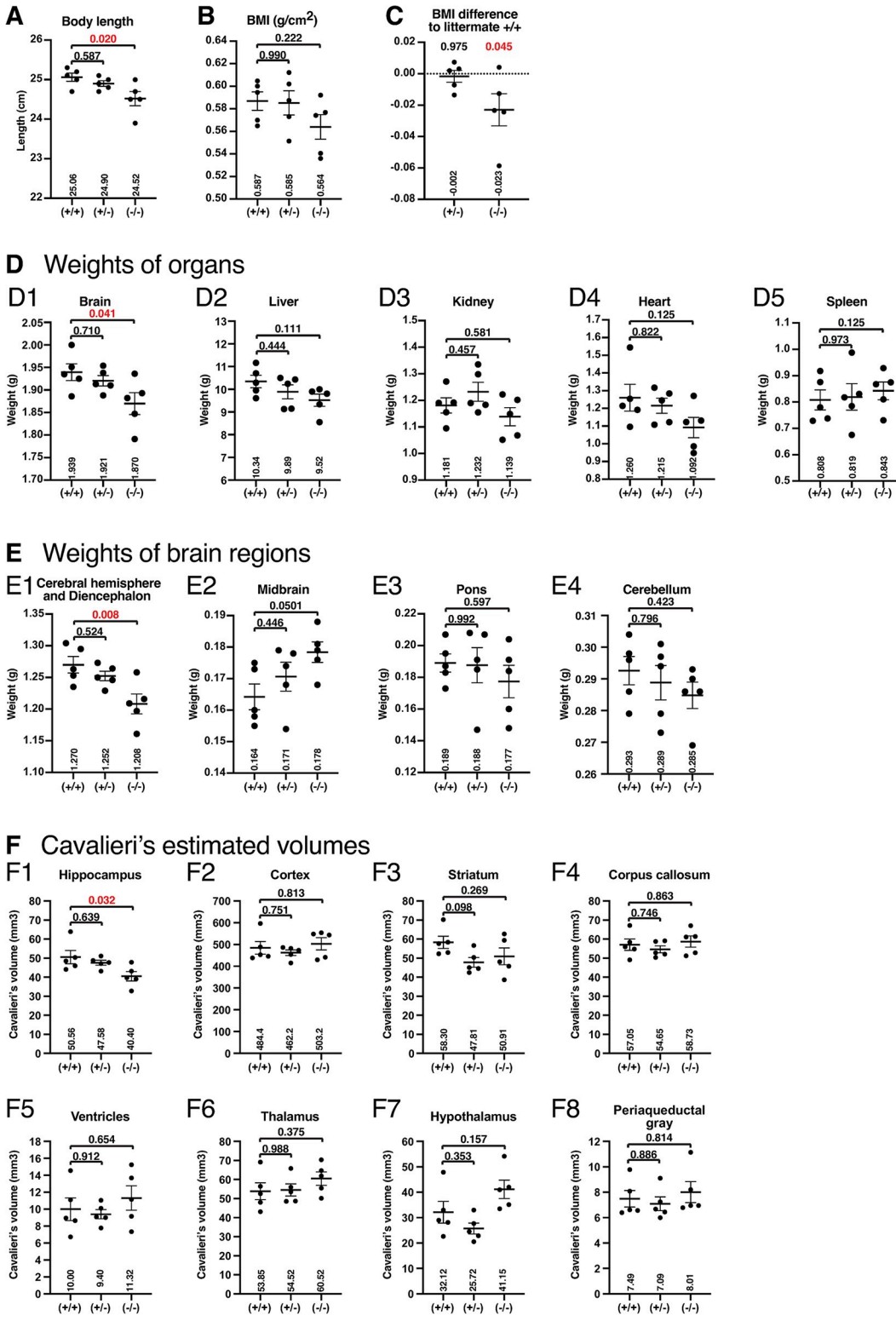

**Fig 8. Reduced brain weight and hippocampal volume in *Slc22a23* knockout rats.** (A) Body length (from nose to anus) at 15 weeks of age in the 2023 summer experiment (male, n = 5). (B) Body mass index (BMI). The BMI was calculated using the following formula: BMI = body weight (grams)/ nose-to-anus length$^2$ (cm$^2$) (C) Difference in the BMI compared with corresponding +/+ littermates. (D) The weight of organs at 15 weeks of age. The brain (D1), liver (D2), kidney (D3), heart (D4), and spleen (D5) were isolated and weighed. (E) The brains isolated in D1 were divided into four regions: cerebral

hemisphere and diencephalon (E1), midbrain (E2), pons (E3), and cerebellum (E4). (F) The T2-weighted magnetic resonance images of the rat brains at 14 weeks of age were analyzed. According to the rat brain atlas, the volumes of eight brain regions (F1 –F8) were estimated using the Cavalieri estimator function of Stereo Investigator. The error bars show the mean ± SEM. The p-values were calculated using Dunnett's multiple comparisons with control +/+ rats.

considered poor [30]. The *Slc22a23* gene transcript is increased by fetal undernutrition (**Fig 1**), suggesting that *Slc22a23* is an important gene for adaptation to a nutrient-poor environment. The *Slc22a23* knockout rats were not overweight and showed no lethal consequences when fed *ad libitum* in our animal facility (**Fig 3**). In contrast, the *Slc22a23* knockout rats were fit and performed better in slow muscle endurance tests than the *Slc22a23*-proficient rats (**Fig 7**). Because the *Slc22a23* gene may have been acquired to adapt to nutrient-poor conditions, *Slc22a23* deficiency may have some disadvantages, especially in nutrient-restricted (starved) conditions. The *Slc22a23* knockout rats generated in this study could be a useful animal model to elucidate the pathogenic mechanisms of lifestyle-related NCDs and psychiatric disorders prevalent in our modern nutrient-rich environment.

## A discrepancy in the phenotype

In our previous study [31], we demonstrated that undernutrition during fetal development induces a hyperactive and anti-anxiety phenotype in rats. In the current study, we identified that undernutrition during fetal development leads to irreversible overexpression of the *Slc22a23* gene in the rat brain (**Fig 1**). We then generated *Slc22a23* knockout rats (**Fig 3**), which unexpectedly exhibited a hyperactive locomotor phenotype (**Fig 5A**) with no effect on the anxiety/anti-anxiety phenotype (**Fig 5A5** **and** **S4 Fig**). These findings suggest that the *Slc22a23* gene alone cannot fully explain the molecular mechanism underlying the DOHaD-associated phenotype induced by fetal undernutrition.

## Overexpression of the *Slc22a23* gene

Undernutrition during fetal development induces an irreversible increase of *Slc22a23* gene expression (**Fig 1**), which may lead to excessive nutrient uptake when the environment becomes nutrient-rich after birth. Overexpression of the *Slc22a23* gene in a nutrient-rich environment may be involved in the development of lifestyle-related NCDs and psychiatric disorders observed in the DOHaD phenotype. Therefore, we carried out this study assuming that the *Slc22a23* gene would be the key gene for the understanding of DOHaD-associated disorders. Several reports have suggested that an increase in *Slc22a23* gene expression is involved in various diseases. DNA demethylation occurs at the *Slc22a23* gene locus in endometrial stromal cells [32], and the *Slc22a23* gene transcript is increased in the endometrial stromal cells [32] and laryngeal squamous cell carcinoma [33]. The single nucleotide polymorphism (SNP) associated with endometriosis patients is located in the 3'-untranslated region of the *Slc22a23* gene, which is a predicted target site for microRNA binding and may participate in downregulation of *Slc22a23* gene expression [34]. Genome-wide association studies found SNPs associated with various conditions, including inflammatory bowel disease [35–37] and QT prolongation in patients taking an antipsychotic drug (quetiapine) [38]. Although these SNPs are located within the introns of the *Slc22a23* gene or are synonymous mutations in the exon, SNPs in non-coding regions have received increased attention due to their potential role in human disease [39]. These findings, including ours, suggest that increased *Slc22a23* gene expression is involved in the pathogenesis of several diseases. However, the current study is limited in elucidating the pathophysiology of the *Slc22a23* gene overexpression because

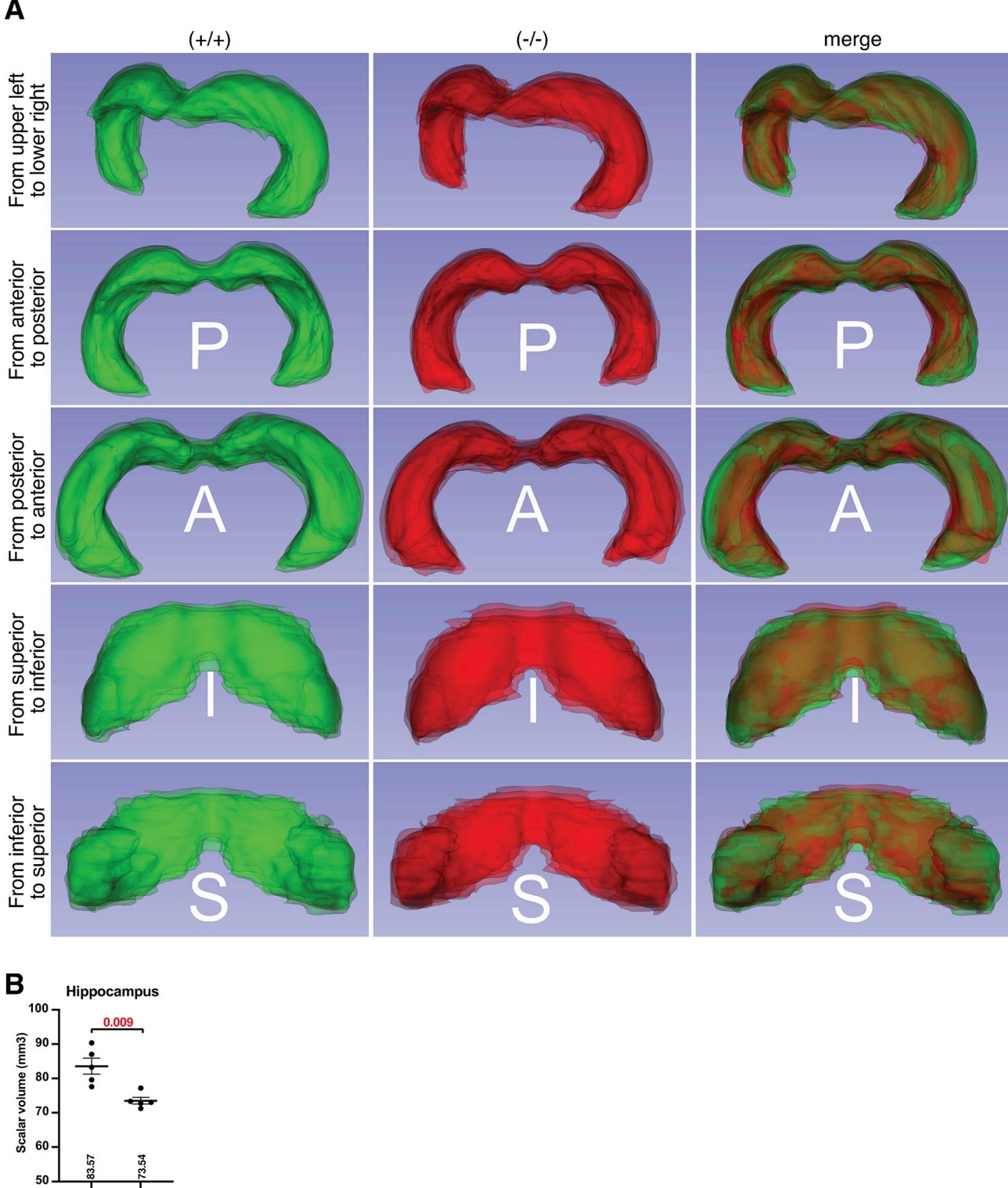

**Fig 9. Reduced volume of the hippocampus in *Slc22a23* knockout rats in a three-dimensional model.** (A) A three-dimensional surface representation of the hippocampus. The three-dimensional models were constructed and superimposed on each other in each genotype (n = 5 in each genotype). Merged images of *Slc22a23*$^{+/+}$ (green) and *Slc22a23*$^{-/-}$ (red) are shown in the right panels. (B) The scalar volume of the three-dimensional hippocampal models (n = 5 in each genotype). The error bars show the mean ± SEM. The p-values show the results of a two-tailed *t*-test (two-sample unequal variance).

*Slc22a23* knockout rats were used instead of *Slc22a23* overexpressing transgenic rats in this study.

## SLC22A23 transporter in neurons

Previous results have demonstrated high *Slc22a23* gene expression in the brain and that its mRNA is expressed in primary neuron cultures [8, 9]. In this study, we showed that *Slc22a23* knockout rats had altered patterns of novel object recognition (**Fig 5B**) and reduced brain weight (**Fig 8D and 8E**) and hippocampal volume (**Figs 8F and 9B**). The neural circuit for novel object recognition in rats involves the hippocampus [40]. The hippocampus in rodents contains 122 hippocampal neuron types based on their axonal and dendritic morphologies, main neurotransmitters, membrane biophysics, and molecular expression [41, 42]. Our findings suggest that the *Slc22a23* gene is particularly important for hippocampal neuron function. Neurons must take up small chemicals, including neurotransmitters, to function properly. The SLC membrane transporter family, with 456 members, is the second largest family of membrane proteins in the human genome but is relatively understudied and comprises a "sparse zone" of our knowledge [43]. However, several SLC membrane transporters are known to be involved in essential functions in neurons, with some members functioning on plasma membranes at synapses. For example, SLC6 family transporters, such as SLC6A2 (noradrenaline), SLC6A3 (dopamine), and SLC6A4 (serotonin), transport monoamine neurotransmitters back into presynaptic neurons [44]. In contrast, some SLC family members function on organellar membranes inside neurons. For example, some SLC transporters participate in the uptake of neurotransmitters into vesicles in the vesicular transport system, including SLC17A6, SLC17A7, and SLC17A8 (glutamate), SLC18A1 and SLC18A2 (monoamines such as dopamine and serotonin), SLC18A3 (acetylcholine), and SLC32A1 (gamma-aminobutyric acid and glycine) [45]. In a future study, we would like to analyze the plasma samples collected from $Slc22a23^{+/+}$, $Slc22a23^{+/-}$, and $Slc22a23^{-/-}$ rats to identify the SLC22A23 transporter substrates. We also would like to immunostain the hippocampus using anti-SLC22A23 antibodies to determine the types of neurons that express the SLC22A23 transporter and the localization of SLC22A23 transporters in each neuron.

## Supporting information

**S1 Fig. The behavioral and phenotyping experiments of rats carried out in this study.** (A) 2022 winter-1, (B) 2022 winter-2, (C) 2023 summer. Limited resources were the primary reason for the choice of the sample size in this study [46] because we had limited breeding space available and had to use gender and litter-matched groups of $Slc22a23^{+/+}$, $Slc22a23^{+/-}$, $Slc22a23^{-/-}$ rats in the experiment. The expected effect size was not available before the experiment started because the phenotypic analysis of *Slc22a23* knockout rats was novel. A post-hoc power analysis was performed using the statistical software G*Power (version 3.1) [47] with the following settings (Test family: F tests, Statistical test: ANOVA, Type of power analysis: Post hoc, Effect size: 0.5, alpha error probability: 0.05, Number of groups: 3). The calculated values of the power (1-beta error probability) were 0.31 (when the total sample size is 15), 0.46 (when the total sample size is 21) and 0.52 (when the total sample size is 24). The power would be 0.77 (total sample size 39) if we combined two datasets (e.g. the results of 8 weeks in 2022 winter-1 and 2023 summer).
(PDF)

**S2 Fig. Comparisons of weight gain, food intake, and water intake.** After being housed in a trio ($Slc22a23^{+/+}$, $Slc22a23^{+/-}$, and $Slc22a23^{-/-}$), each rat was individually housed in a cage

from postnatal day (P)33 to P63 to measure food and water intake. Their (A) body weight, (B) food intake, and (C) water intake were measured every 3 days. The winter-2022-2 results (male, n = 5 in each group) are shown.
(PDF)

**S3 Fig. Increased total distance travelled in *Slc22a23* knockout rats.** The results of the 2023 summer experiment (male: n = 5 in each group, female: n = 7 in each group) are shown as in Fig 5A. The factorial ANOVA of the data in S3 Fig can be found in Panel B of S2 Table.
(PDF)

**S4 Fig. *Slc22a23* knockout rats exhibited no significant difference in the elevated plus-maze test.** The apparatus had four arms (50 cm length, 10 cm width, 51.5 cm height); two of the arms lacked walls (open arms) and the other two arms had walls (closed arms). At the start of the test, the test rat was placed in the central zone of the apparatus. The movement of the test rat was recorded using a CompACT video tracking system (Muromachi Kikai) for 5 minutes. The width of the border was set to 5 cm. The time spent in each arm and the number of entries into each arm were recorded. The 2022 winter-1 results (male: n = 8 in each group) are shown. The error bars show the mean ± SEM. The p-values were calculated using Dunnett's multiple comparisons with control +/+ rats. The factorial ANOVA of the data in S4 Fig can be found in Panel C of S2 Table.
(PDF)

**S5 Fig. *Slc22a23* knockout rats showed reduced brain weight.** The results are shown as in Fig 8D, but the organs were isolated from 2022 winter-1 (male, n = 8, high-fat diet) and 2022 winter-2 (male, n = 5, normal chow) animals at 16 weeks of age. The error bars show the mean ± SEM. The p-values were calculated using Dunnett's multiple comparisons with control +/+ rats. The factorial ANOVA of the data in S5 Fig can be found in Panel E of S3 Table.
(PDF)

**S1 Table. The full dataset of the microarray analysis in Fig 1A.**
(XLS)

**S2 Table. Factorial ANOVA of behavioral tests.** Values below each genotype are mean ± SEM. Factorial ANOVA (with repeated measurements) was performed with (A) 'Genotype' as a between-subjects factor, and 'Age' as a within-subjects factor; (B) 'Genotype' and 'Gender' as between-subjects factors, and 'Age' as a within-subjects factor; (C) 'Genotype' as a between-subjects factor, and 'Age' as a within-subjects factor; (D) 'Genotype' as a between-subjects factor, and 'Age' and 'Day' as within-subjects factors; (E) 'Genotype' as a between-subjects factor, and 'Age' and 'Intruder' as within-subjects factors.
(PDF)

**S3 Table. Factorial ANOVA of physical assessments.** Values below each genotype are mean ± SEM. Factorial ANOVA (with repeated measurements) was performed with (A) 'Genotype' as a between-subjects factor, and 'Age' as a within-subjects factor; (B) 'Genotype' and 'Gender' as between-subjects factors, and 'Age' as a within-subjects factor; (C) 'Genotype' as a between-subjects factor, and 'Age' as a within-subjects factor. In (D) and (E), factorial ANOVA (without repeated measurements) was performed.
(PDF)

**S1 Raw images. All the original blot and gel images in the manuscript's main figures and supplementary figures.**
(PDF)

## Acknowledgments

We would like to thank Dr. Hirotaka Iwasaki for providing the C2C12 cells. We also would like to thank Mr. Kazuma Fujita and Mr. Kenta Wakitani for their help in sample collection. We thank Lisa Kreiner, PhD, from Edanz (https://jp.edanz.com/ac) for editing the English text of a draft of this manuscript.

## Author Contributions

**Conceptualization:** Yasuhiro Uchimura, Jun Udagawa.

**Data curation:** Yasuhiro Uchimura.

**Formal analysis:** Yasuhiro Uchimura.

**Funding acquisition:** Yasuhiro Uchimura, Yataro Daigo, Tomoji Mashimo.

**Investigation:** Yasuhiro Uchimura, Kodai Hino, Kosuke Hattori, Yoshinori Kubo, Airi Owada, Tomoko Kimura, Lucia Sugawara, Shinji Kume, Jean-Pierre Bellier, Daijiro Yanagisawa, Akihiko Shiino, Takahisa Nakayama, Yataro Daigo, Tomoji Mashimo, Jun Udagawa.

**Methodology:** Yasuhiro Uchimura, Kosuke Hattori, Lucia Sugawara, Jean-Pierre Bellier, Daijiro Yanagisawa, Akihiko Shiino, Takahisa Nakayama, Yataro Daigo, Tomoji Mashimo.

**Project administration:** Yasuhiro Uchimura.

**Resources:** Yasuhiro Uchimura, Kosuke Hattori, Lucia Sugawara, Shinji Kume, Jean-Pierre Bellier, Daijiro Yanagisawa, Akihiko Shiino, Takahisa Nakayama, Yataro Daigo, Tomoji Mashimo.

**Supervision:** Yasuhiro Uchimura, Shinji Kume, Yataro Daigo, Tomoji Mashimo, Jun Udagawa.

**Validation:** Yasuhiro Uchimura.

**Visualization:** Yasuhiro Uchimura.

**Writing – original draft:** Yasuhiro Uchimura.

**Writing – review & editing:** Yasuhiro Uchimura, Jun Udagawa.

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
