## [Decision Letter · Decision Letter 0]

13 Jun 2024

PONE-D-24-12532Knockout of the orphan membrane transporter Slc22a23 leads to a lean and hyperactive phenotype with a small hippocampal volumePLOS ONE

Dear Dr. Uchimura,

Thank you for submitting your manuscript to PLOS ONE. After careful consideration, we feel that it has merit but does not fully meet PLOS ONE’s publication criteria as it currently stands. Therefore, we invite you to submit a revised version of the manuscript that addresses the points raised during the review process.

We look forward to receiving your revised manuscript.

Kind regards,

Takahiro Nemoto, Ph.D

Academic Editor

PLOS ONE

Journal Requirements:

2. To comply with PLOS ONE submissions requirements, in your Methods section, please provide additional information regarding the experiments involving animals and ensure you have included details on (a) methods of sacrifice, (b) methods of anesthesia and/or analgesia, and (c) efforts to alleviate suffering.

This work was supported by Japan Society for the Promotion of Science (JSPS; https://www.jsps.go.jp/) - JP 19K08274 [YU], JP 16H06277 / JP 22H04923 [YD], and JP 16H06276 [TM]. The funders had no role in the study design, data collection and analysis, decision to publish, or manuscript preparation.

Additional Editor Comments:

The manuscript has been rigorously reviewed by two peer reviewers, who have pointed out some areas for improvement. The editors will ask the authors to make these corrections.

Reviewers' comments:

Reviewer's Responses to Questions

**Comments to the Author**

1. Is the manuscript technically sound, and do the data support the conclusions?

Reviewer #1: Yes

Reviewer #2: Yes

2. Has the statistical analysis been performed appropriately and rigorously? 

Reviewer #1: No

Reviewer #2: Yes

3. Have the authors made all data underlying the findings in their manuscript fully available?

Reviewer #1: No

Reviewer #2: Yes

4. Is the manuscript presented in an intelligible fashion and written in standard English?

Reviewer #1: Yes

Reviewer #2: Yes

5. Review Comments to the Author

**Reviewer #1: **This study shows that uplegulation of Slc22a23 mRNA in the rat brain after undernutrition during fetal development. Then, the authors generated Slc22a23 knockout rats and found the KO rats showed lean, hyperactive, and reduced hippocampal volume. These results are interesting. I have some requests.

1. If transcript of a Slc22a23 gene is important after undernutrition during fetal development, overexpression of Slc22a23 should have critical effects to pathophysiology. The authors should show this evidence, or at least discuss about it.

2. The authors should describe all F values, degrees of freedom, exact p values of main effects of genotype, sex, or age, and their interactions after two- or three-way ANOVA with repeated measure in supplementary table.

3. The authors should show representable MRI 3D images of KO and WT rat brains.

4. ‘Written consent’ is not necessary.

**Reviewer #2: **Dear author;

Your work has been prepared with a very original and sufficient methodology.

As a suggestion, the concentrations of antibodies examined in immunoblotting can also be presented. In addition, it would be useful to clearly state how the bands in the membranes obtained for western blot were calculated and which statistical method was used.

Best regards

6. PLOS authors have the option to publish the peer review history of their article (what does this mean?). If published, this will include your full peer review and any attached files.

Reviewer #1: No

Reviewer #2: No

---

## [Author Response · Author response to Decision Letter 0]

29 Jul 2024

Reviewer #1: This study shows that uplegulation of Slc22a23 mRNA in the rat brain after undernutrition during fetal development. Then, the authors generated Slc22a23 knockout rats and found the KO rats showed lean, hyperactive, and reduced hippocampal volume. These results are interesting. I have some requests.

1. If transcript of a Slc22a23 gene is important after undernutrition during fetal development, overexpression of Slc22a23 should have critical effects to pathophysiology. The authors should show this evidence, or at least discuss about it.

Thank you very much for pointing this out. We have a section entitled “Overexpression of the Slc22a23 gene” in the DISCUSSION section.

2. The authors should describe all F values, degrees of freedom, exact p values of main effects of genotype, sex, or age, and their interactions after two- or three-way ANOVA with repeated measure in supplementary table.

Thank you very much for pointing this out. We have added these data to the S2 and S3 Tables. 

3. The authors should show representable MRI 3D images of KO and WT rat brains.

Thank you very much for pointing this out. We have added the 3D images of MRI to the Fig 9. I realised my mistake of using the wrong scale factor in the initial analysis of Stereo Investigator Fig 8F. I corrected the value in Fig 8F.

4. ‘Written consent’ is not necessary.

I removed them from the manuscript.

Reviewer #2: Dear author;

Your work has been prepared with a very original and sufficient methodology.

As a suggestion, the concentrations of antibodies examined in immunoblotting can also be presented. 

I added the concentrations of antibodies in the sections of MATERIALS AND METHODS.

In addition, it would be useful to clearly state how the bands in the membranes obtained for western blot were calculated and which statistical method was used.

I added the value in Fig3D and then added the following sentences in the sections of MATERIALS AND METHODS; The signals on the membrane were detected using a chemiluminescence imaging system (FUSION SOLO.6S.EDGE; Viber-Lourmat, Collégien, France) and saved in TIF format. The TIF files were opened in ImageJ (version 1.54g; NIH, US). The band intensities of the SLC22A23 protein, indicated by the arrowhead, were measured using the Analyze function in ImageJ. The band intensities of the SLC22A23 protein were compensated by the band intensities of the loading controls (anti-Tubulin betaIII, anti-GFAP or anti-CNPase). The relative band intensities of the SLC22A23 protein compared to +/+ are shown in Fig 3D.

---

## [Decision Letter · Decision Letter 1]

7 Aug 2024

PONE-D-24-12532R1Knockout of the orphan membrane transporter Slc22a23 leads to a lean and hyperactive phenotype with a small hippocampal volumePLOS ONE

Dear Dr. Uchimura,

Thank you for submitting your manuscript to PLOS ONE. After careful consideration, we feel that it has merit but does not fully meet PLOS ONE’s publication criteria as it currently stands. Therefore, we invite you to submit a revised version of the manuscript that addresses the points raised during the review process.

We look forward to receiving your revised manuscript.

Kind regards,

Takahiro Nemoto, Ph.D

Academic Editor

PLOS ONE

Journal Requirements:

Additional Editor Comments:

The manuscript has been reviewed again impartially by two reviewers. The content of the paper has been improved since the first draft, but it seems to have the potential to be even better. The editors hope that the authors will revise the paper in accordance with the reviewers' suggestions and resubmit it.

Reviewers' comments:

Reviewer's Responses to Questions

**Comments to the Author**

1. If the authors have adequately addressed your comments raised in a previous round of review and you feel that this manuscript is now acceptable for publication, you may indicate that here to bypass the “Comments to the Author” section, enter your conflict of interest statement in the “Confidential to Editor” section, and submit your "Accept" recommendation.

Reviewer #1: (No Response)

Reviewer #2: All comments have been addressed

2. Is the manuscript technically sound, and do the data support the conclusions?

Reviewer #1: Yes

Reviewer #2: Yes

3. Has the statistical analysis been performed appropriately and rigorously? 

Reviewer #1: Yes

Reviewer #2: Yes

4. Have the authors made all data underlying the findings in their manuscript fully available?

Reviewer #1: Yes

Reviewer #2: Yes

5. Is the manuscript presented in an intelligible fashion and written in standard English?

Reviewer #1: Yes

Reviewer #2: Yes

6. Review Comments to the Author

Reviewer #1: The revised manuscript is improved following the reviewers' comments.

One point should be addressed: Regarding the pathophysiology of overexpression of Slc22a23 gene, the authors should describe the limitation of this study using knockout rats instead of transgenic rats.

Reviewer #2: Dear Authors; I believe that your broadcast will be better after the necessary checks have been made. I believe that if the study is published, it will contribute to the field of science and fill the gap in this field.

7. PLOS authors have the option to publish the peer review history of their article (what does this mean?). If published, this will include your full peer review and any attached files.

Reviewer #1: **Yes: **Takatoshi Hikida

Reviewer #2: **Yes: **Musa TATAR

---

## [Author Response · Author response to Decision Letter 1]

8 Aug 2024

Reviewer #1: The revised manuscript is improved following the reviewers' comments.

One point should be addressed: Regarding the pathophysiology of overexpression of Slc22a23 gene, the authors should describe the limitation of this study using knockout rats instead of transgenic rats.

Thank you very much for pointing this out. I added the following sentence in the Discussion: However, the current study is limited in elucidating the pathophysiology of the Slc22a23 gene overexpression because Slc22a23 knockout rats were used instead of Slc22a23 overexpressing transgenic rats in this study.

---

## [Decision Letter · Decision Letter 2]

13 Aug 2024

Knockout of the orphan membrane transporter Slc22a23 leads to a lean and hyperactive phenotype with a small hippocampal volume

PONE-D-24-12532R2

Dear Dr. Uchimura,

We’re pleased to inform you that your manuscript has been judged scientifically suitable for publication and will be formally accepted for publication once it meets all outstanding technical requirements.

Kind regards,

Takahiro Nemoto, Ph.D

Academic Editor

PLOS ONE

Additional Editor Comments (optional):

The manuscript has been reviewed by two reviewers and deemed sufficiently revised to be suitable for publication. Congratulations.

Reviewers' comments:

Reviewer's Responses to Questions

**Comments to the Author**

1. If the authors have adequately addressed your comments raised in a previous round of review and you feel that this manuscript is now acceptable for publication, you may indicate that here to bypass the “Comments to the Author” section, enter your conflict of interest statement in the “Confidential to Editor” section, and submit your "Accept" recommendation.

Reviewer #1: All comments have been addressed

2. Is the manuscript technically sound, and do the data support the conclusions?

Reviewer #1: Yes

3. Has the statistical analysis been performed appropriately and rigorously? 

Reviewer #1: Yes

4. Have the authors made all data underlying the findings in their manuscript fully available?

Reviewer #1: Yes

5. Is the manuscript presented in an intelligible fashion and written in standard English?

Reviewer #1: Yes

6. Review Comments to the Author

Reviewer #1: (No Response)

7. PLOS authors have the option to publish the peer review history of their article (what does this mean?). If published, this will include your full peer review and any attached files.

Reviewer #1: **Yes: **Takatoshi Hikida

---

## [Editor Report · Acceptance letter]

19 Aug 2024

PONE-D-24-12532R2 

PLOS ONE

Dear Dr. Uchimura, 

I'm pleased to inform you that your manuscript has been deemed suitable for publication in PLOS ONE. Congratulations! Your manuscript is now being handed over to our production team.

Kind regards, 

on behalf of

Dr. Takahiro Nemoto 

Academic Editor

PLOS ONE